



# The 2020 glacial lake outburst flood at Jinwuco, Tibet: causes, impacts, and implications for hazard and risk assessment

Guoxiong Zheng[1,2,3], Martin Mergili[4,5], Adam Emmer[4], Simon Allen[2,6], Anming Bao[1,7], Hao Guo[8],
Markus Stoffel[2,9,10]

[1]State Key Laboratory of Desert and Oasis Ecology, Xinjiang Institute of Ecology and Geography, Chinese Academy of Sciences, 830011 Urumqi, China
[2]Climatic Change Impacts and Risks in the Anthropocene (C-CIA), Institute for Environmental Sciences, University of Geneva, 1205 Geneva, Switzerland
[3]University of Chinese Academy of Sciences, 100049 Beijing, China
[4]Institute of Geography and Regional Science, University of Graz, 8010 Graz, Austria
[5]Institute of Applied Geology, University of Natural Resources and Life Sciences (BOKU), 1190 Vienna, Austria
[6]Department of Geography, University of Zurich, 8057 Zurich, Switzerland
[7]China-Pakistan Joint Research Center on Earth Sciences, CAS-HEC, 45320 Islamabad, Pakistan
[8]School of Geography and Tourism, Qufu Normal University, 276800 Rizhao, China
[9]Dendrolab.ch, Department of Earth Sciences, University of Geneva, 1205 Geneva, Switzerland
[10]Department of F.A. Forel for Environmental and Aquatic Sciences, University of Geneva, 1205 Geneva, Switzerland

*Correspondence to*: G. Zheng (zhengguoxiong17@mails.ucas.edu.cn); A. Bao (baoam@ms.xjb.ac.cn)





**Abstract.** We analyze and reconstruct a recent Glacial Lake Outburst Flood (GLOF) process chain on 26 June 2020, involving the moraine-dammed proglacial lake Jinwuco (30.356°N, 93.631°E) in eastern Nyainqentanglha, Tibet, China. Satellite images reveal that from 1965 to 2020, the surface area of Jinwuco has expanded by 0.2 km$^2$ (+56%) to 0.56 km$^2$, and subsequently decreased to 0.26 km$^2$ (–54%) after the GLOF. Estimates based on topographic reconstruction and sets of published empirical relationships indicate that the GLOF had a volume of 10 million m$^3$, an average breach time of 0.62 hours, and an average peak discharge of 5,390 m$^3$/s at the dam. Based on pre- and post-event high-resolution satellite scenes, we identified a large progressive debris landslide originating from western lateral moraine, having occurred 5–17 days before the GLOF. This landslide was most likely triggered by extremely heavy, south Asian monsoon-associated rainfall in June. The time lag between the landslide and the GLOF suggests that pre-weakening of the dam due to landslide-induced outflow pushed the system towards a tipping point, that was finally exceeded following subsequent rainfall, snowmelt, a secondary landslide, or calving of ice into the lake. We back-calculate part of the GLOF process chain, using the GIS-based open source numerical simulation tool r.avaflow. Two scenarios are considered, assuming a debris landslide-induced impact wave with overtopping and resulting retrogressive erosion of the moraine dam (Scenario A), and retrogressive erosion due to pre-weakening of the dam without a major impact wave (Scenario B). Both scenarios yield plausible results which are in line with empirically derived ranges of peak discharge and breach time. The breaching process is characterized by a slower onset and a resulting delay in Scenario B, compared to Scenario A. Evidence, however, points towards Scenario B as a more realistic possibility. There were no casualties from this GLOF but it caused severe destruction of infrastructure (e.g. roads and bridges) and property losses in downstream areas. Given the clear role of continued glacial retreat in destabilizing the adjacent lateral moraine slopes, and directly enabling the landslide to deposit into the expanding lake body, the GLOF process chain under Scenario B can be robustly attributable to anthropogenic climate change, while downstream consequences have been enhanced by the development of infrastructure on exposed flood plains. Such process chains could become more frequent under a warmer and wetter future climate, calling for comprehensive and forward-looking risk reduction planning.

## 1 Introduction

The widespread retreat and thinning of glaciers observed in the 20[th] century has, in most regions, accelerated over recent decades as a consequence of global warming (Hock et al., 2019; Zemp et al., 2019), leading to the rapid expansion of glacial lakes (Shugar et al., 2020; Wang et al., 2020). When water is suddenly and catastrophically released, Glacial Lake Outburst Floods (GLOFs) can cause severe societal and geomorphic impacts over large distances downstream (Lliboutry, 1977; Carrivick and Tweed, 2016; Cook et al., 2018). This threat is most apparent in high-mountain Asia, where severe, high-magnitude GLOFs have been recorded (Harrison et al., 2018; Nie et al., 2018; Veh et al., 2109; Zheng et al., 2020). Lakes are increasing rapidly in both number and area (Gardelle et al., 2011; Zhang et al., 2015; Zheng et al., 2019), and GLOF impacts can extend across national boundaries to create severe challenges for early warning or other risk reduction strategies (Khanal et al., 2015; Allen et al., 2019). Scientific attention has primarily focused upon moraine-dammed glacial lakes, owing to their large volumes (Fujita et al., 2013; Veh et al., 2020), weak dam composition, and predisposition to various triggering mechanisms, of which ice and/or rock avalanches are most commonly reported (Emmer and Cochachin, 2013; Nie et al., 2018). However, despite numerous large scale GLOF hazard and risk studies having been conducted at national (e.g. Wang et al., 2015; Rounce et al., 2017; Dubey and Goyal, 2020) and larger regional scales (e.g. Ives et al., 2010; Schwanghart et al., 2016), there remains a relative lack of site-specific investigations and reconstructions of past GLOF disasters. In fact, only a small fraction of recorded moraine-dammed GLOF events across high-mountain Asia have been comprehensively studied (e.g. Narama et al., 2010; Allen et al., 2016; Gurung et al., 2017; Byers et al., 2018; Nie et al., 2020), and as such, triggering mechanisms and processes for the majority of cases remain speculative. This is particularly true for the large number of GLOF



events that have occurred during the monsoon months (Richardson and Reynolds, 2000), during which time a lack of cloud-free satellite image and/or safe access to the field can prevent detailed assessment. Nonetheless, based on limited case evidence from the Himalaya (Allen et al., 2016) and elsewhere (e.g. Clague and Evans, 2000; Worni et al., 2012), it is reasonable to assume that heavy rainfall may be an important, yet often underrated trigger of GLOF events and related process chains. In view of projected climate change bringing warmer and also wetter conditions to monsoon-affected parts of high-mountain

Asia (Mathison et al., 2013; Kitoh, 2017; Sanjay et al., 2017), there is hence an urgent need to learn from recent GLOF disasters, in order to improve our ability to assess and eventually manage future risks. This includes consideration of not only the physical drivers of GLOF hazard but also the underlying dimensions of societal exposure and vulnerability that may ultimately drive GLOF risk (Huggel et al., 2020).

Here, we aim to provide a rapid yet comprehensive investigation of a recent GLOF disaster occurring in eastern

Nyainqentanglha, Tibet, China, during the onset of the 2020 monsoon, combining numerical modelling with the analysis of remotely sensed data, eye-witness accounts and media reports. Specifically, the objectives of this study are to reconstruct both the longer-term conditioning and short-term triggering and dynamics of the GLOF event and to assess the immediate impacts upon downstream communities. Ultimately, we explore the extent to which this GLOF disaster can be attributed to anthropogenic climate change or other drivers, and discuss broader implications for future GLOF hazard and risk assessment.

**2 Study area**

The study area is located at the Nidou Zangbo ("Zangbo" refers to rivers in Tibetan) watershed in the eastern Nyainqentanglha, Tibet Autonomous Region, China (Fig. 1). The Nidou Zangbo originates from the eastern foothills of the Luola Mountains near the north of Zhongyu township in Lhari County with a length of 68 km (Liu, 2014) and is the right tributary of Yi'ong Zangbo. The watershed covers an area of 1,267 $km^2$ with an elevation difference of 3,715 m. The topography of the watershed

is high in the west and low in the east, with an average elevation of >5,000 m. Most of the peaks in the watershed reach more than 6,000 m, with the highest peak, Nenang Peak, at 6,870 m. The glacial and paraglacial landscapes are highly developed in the watershed, with a total of 294 glaciers covering a combined area of 396 $km^2$ (RGI Consortium, 2017). There are 96 glacial lakes (≥900 $m^2$) distributed in the watershed with a total area of 3.2 $km^2$ and an average area of 0.03 $km^2$ (Zheng et al., 2020), of which the majority are moraine-dammed lakes. The watershed is part of the sub-humid monsoon climate zone of the plateau

with cold winters and cool summers (Sun et al., 2014). The average annual temperature recorded from the Lhari weather station during 1980–2019 is –0.2°C with an average lowest temperature of –10.6°C in January and average highest temperature of 8.9°C in July (Fig. 1). The average annual total precipitation is 753 mm and about 80% is concentrated from May to September (Fig. 1), largely controlled by the south Asian monsoon (Sun et al., 2014).



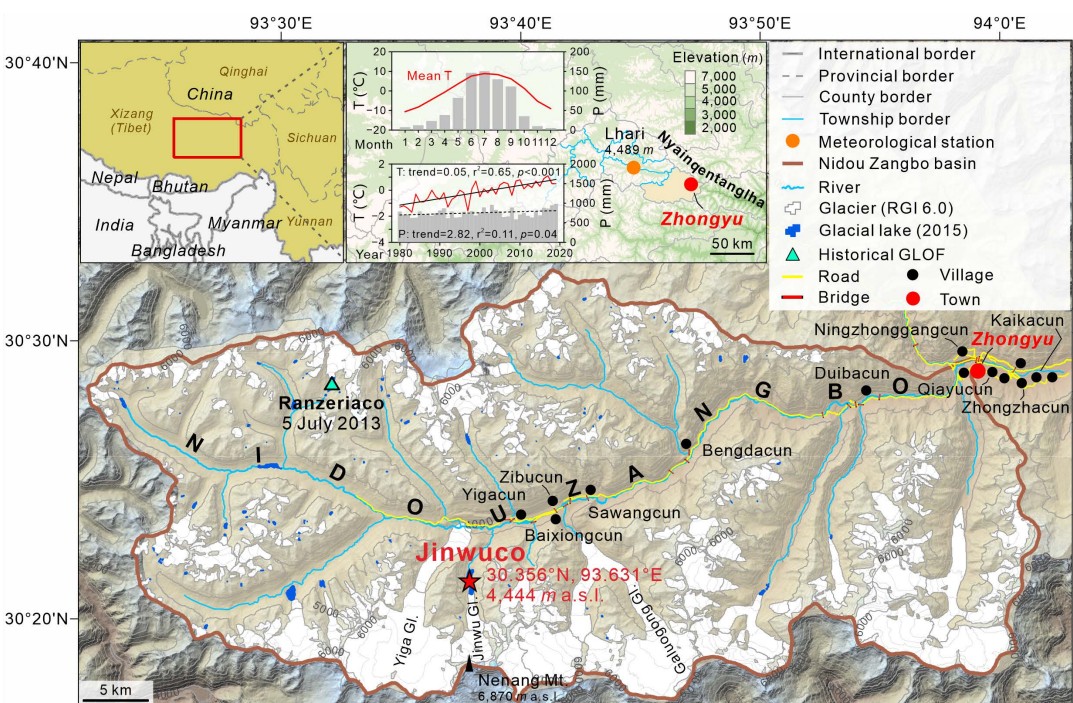

**Figure 1. Location of Jinwuco and surrounding geographic features, including terrain, rivers, glaciers, glacial lakes, settlements and major infrastructures (roads and bridges) downstream. The climate statistics in the upper middle panel are based on the Lhari weather station, approx. 50 km away from Jinwuco, 1980–2019. Topographic base map is based on the DEM generated from Ziyuan-3 stereoscopic pairs. The borders and location of the villages are from © Baidu Maps.**

Jinwuco (also known as Jiwengco, 30.356°N, 93.631°E) is a proglacial moraine-dammed lake (Fig. 2) which is also the largest glacial lake within the Nidou Zangbo basin (Zheng et al., 2020). It is situated at the terminal of the Jinwu Glacier that covers an area of 7.9 km$^2$ and the lake is in contact with the steep ice tongue. The ice tongue enters the lake about 350 m below the steep ice cliff that has a height of approx. 250 m and an average slope of 35° measured based on the satellite scene and DEM acquired on 25 August 2020 (Table 1). The main axis of the lake is oriented in a north-south direction, flanked by steep slopes and lateral moraines. The average height of the lateral moraines is over 100 m and the slopes are about 40°. The upper end and both lateral moraine slopes surrounding the lake are seasonally snow-covered, with an ablation period from April to September and an accumulation period from October to March. The lake shows a typical elongated irregular oblong shape with a straight-line length of approx. 1.8 km and its width varies in different positions. Before the GLOF of 26 June 2020, it exhibited wider front and rear sections and a narrow middle section, with the section near the dam being 0.33 km wide. The part near the glacier is 0.24 km and the middle section is about 0.23 km wide. The dam of the lake is formed by a vegetation-covered terminal moraine, and has a length of about 400 m perpendicular to the streamline with an average slope of the distal (outer) face of 20° and an average height of 10 m. The outlet channel is located on the western side of the moraine dam where there is a permanent surficial outflow (i.e. dam freeboard = 0 m).

From the administrative location, the Zhongyu township where Jinwuco is situated is part of the southeastern Lhari County, 118 km from the county seat. It contains 14 administrative villages in total. Six of them are within the middle and upper reaches of the Nidou Zangbo basin (Fig. 1).





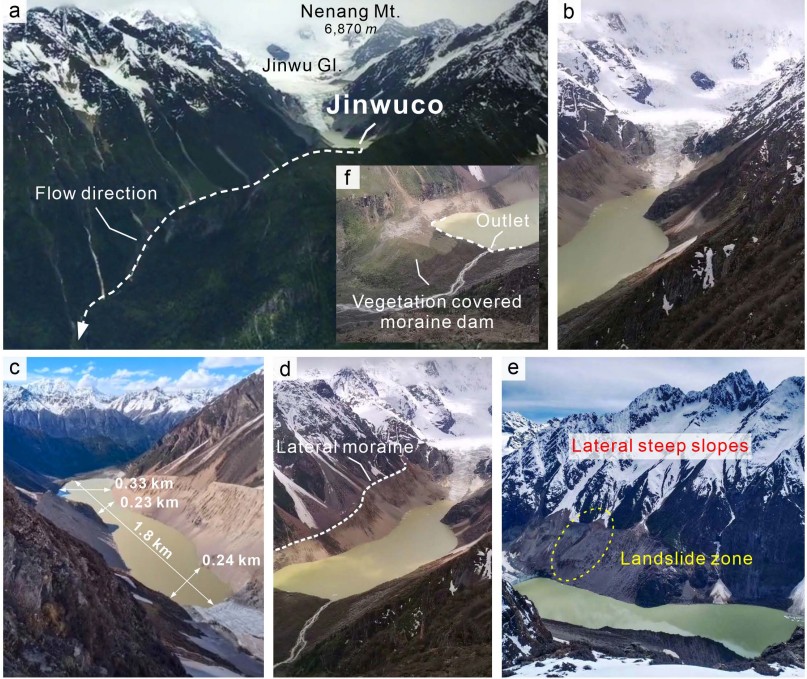

**Figure 2. Overview of Jinwuco before the collapse of the dam. (a) An overall view of Jinwuco and surroundings taken from the opposite side. (b) Crevassed hanging glacier ice terminated in Jinwuco, taken from front to back. (c) Morphological parameters of Jinwuco, photo taken from back to front. (d) Lateral moraine and steep mountain slopes from right to left and (e) from left to right. (f) Vegetation covered moraine dam and the position of the outlet. Photos: Q. Quying (used with permission).**

## 3 Materials and methods

### 3.1 Long-term evolution of Jinwuco and adjacent slopes

To investigate the evolution of Jinwuco, multiple sets of satellite scenes are used to obtain its surface area between 1965 and 2020. The details and parameters of the datasets used are presented in Table 1. The declassified spy photographs of Corona KH-4A and Hexagon KH-9, captured on 31 December 1965 and on 4 January 1976, are first geometrically corrected using a Landsat 8 scene taken on 28 September 2013 as the master image, and are then projected onto the 1984 WGS UTM zone N46 coordinate system and resampled to 3 m and 6 m, respectively. The multispectral and panchromatic bands of Landsat-7 ETM+ and Landsat-8 OLI are fused using a nearest neighbour diffusion-based pan-sharpening algorithm (Sun et al., 2014) to improve the spatial resolution to 15 m.

The outlines of Jinwuco are manually delineated based on these satellite images. The lake area is then calculated based on the UTM projection and its uncertainty is estimated using the following equation (Hanshaw and Bookhagen, 2014):

$$\delta = \frac{P}{G} \times \frac{G^2}{2} \times 0.6872$$

where $P$ is the perimeter of the glacial lake and $G$ is the spatial resolution of the images used.

The regression analysis method is used to quantify the long-term evolution trend of Jinwuco.





**Table 1. Details and parameters of datasets used.**

| Data | Date | Resolution (m) | Purpose | Source |
|---|---|---|---|---|
| Corona KH-4A | 31.12.1965 | 3 | Glacial lake mapping | |
| Hexagon KH-9 | 04.01.1976 | 6 | Glacial lake mapping | |
| Landsat 1-3 MSS | 09.01.1974; 08.06.1978; 12.07.1980 | 60 | Glacial lake mapping | |
| Landsat-5 TM | 23.10.1987; 09.10.1988; 13.11.1989; 25.06.1990; 03.11.1991; 14.06.1992; 01.06.1993; 10.10.1994; 25.07.1995; 15.10.1996; 03.11.1997; 05.10.1998; 22.09.1999; 04.06.2000; 10.13.2001; 13.08.2002; 04.11.2003; 21.10.2004; 06.09.2005; 12.11.2006; 07.05.2007; 19.10.2009; 06.10.2010; 25.11.2011 | 30 | Glacial lake mapping | 1 |
| Landsat-7 EMT+ | 05.08.2008; 16.10.2008; 04.11.2012; 13.06.2012; 26.09.2015 | 30/15* | Glacial lake mapping | |
| Landsat-8 OLI | 28.09.2013; 01.10.2014; 19.08.2016; 10.05.2020 | 30/15* | Glacial lake mapping | |
| Sentinel-2 A&B | 26.09.2017; 11.09.2018; 26.09.2019; 27.07.2020 | 10^ | Glacial lake mapping | 2 |
| Ziyuan-3 02 | 25.08.2020 | 2.1/5.8* | Glacial lake mapping | 3 |
| Gaofen-2 | 07.12.2017 | 0.8/3.2* | GLOF exposure analysis | 3 |
| Sentinel-1 A | 09.06.2020; 21.06.2020; 03.07.2020 | 20 | Dynamic analysis of changes in and around glacial lake | 2 |
| Ziyuan-3 02 digital elevation model (DEM) | 18.12.2017; 25.08.2020 | 3.5 | GLOF process chain simulation | 3 |
| Meteorological station data | 1980–2020 | Daily and annual temperature and precipitation | Regional climate trend and GLOF cause analysis | 4 |

1. USGS – United States Geological Survey (https://earthexplorer.usgs.gov/);
2. ESA – European Space Agency (https://scihub.copernicus.eu/);
3. RSGS – China Remote Sensing Satellite Ground Station – Kashi Ground Station; CRESDA – China Center for Resources Satellite Date and Application (http://www.cresda.com/);
4. CMA – China Meteorological Administration (http://www.cma.gov.cn/).
*Spatial resolution of multispectral/panchromatic image; ^Only multispectral and near-infrared bands (2,3,4,8) with 10 m spatial resolution were used.

## 3.2 Reconstruction and quantification of relevant features

Reconstruction of flood-related features is mainly based on two DEMs (one from 18 December 2017, i.e. before the event, and the other one from 25 August 2020, i.e. after the event) that were generated from Ziyuan-3 stereoscopic pairs (Table 1). A rough DEM of Difference (DoD) is produced from the overlay of the two DEMs. Further, empirical equations estimating lake volume from lake surface area (Section 3.4) are employed. Specifically, the following steps are performed:

1. The lake bottom topography is reconstructed from the post-event DEM and the empirical volume estimation outlined in Section 3.4. In the pre-event DEM, the lake surface elevation is replaced by the reconstructed lake bottom elevation.

2. The landslide scarp area as well as release volume and release thickness distribution are derived from the DoD, also considering the landslide features revealed by the post-event satellite image. The same is done for the landslide deposit.

3. The DoD is used to derive the breach geometry and volume at the moraine dam of Jinwuco.



4. The pre-event and the post-event DEM are mosaicked in order to obtain the best possible quality and the most up-to-date information for each pixel of the study area. This is necessary as – on the one hand – both DEMs show areas of poor quality, with major artefacts. On the other hand, it is necessary to use the post-event DEM for the rear portion of Jinwuco, as the pre-event DEM shows the situation in 2017, which is outdated due to glacier retreat.

As a result, raster maps of the terrain surface (cleared for lake water as well as landslide material), distribution of landslide release and deposition thickness, and distribution of erosion depth at the moraine dam of Jinwuco are available for further analyses. All raster maps are prepared at a cell size of 3 m. The DEM is not corrected for the influence of vegetation, which is considered negligible within the scope of the present work.

Further, Sentinel-1 radar satellite images from 9 June, 21 June, and 3 July 2020 are used to analyse possible changes of the
lake and surroundings in the weeks before the event of June 26, and to relate them to the post-event situation. The use of optical images for this purpose is largely restricted by extensive cloud cover.

### 3.3 Meteorological conditions before the event

In order to ascertain and analyze the meteorological conditions in the region prior to the GLOF, the daily temperature and precipitation data pre- and post-event at the nearest Lhari weather station to Jinwuco are used (Fig. 1). The station is located
approximately 50 km to the northwest of Jinwuco, at a comparable elevation (4,489 vs. 4,444 m asl). The mean and extreme temperature and precipitation over the first half of 2020 are compared to a 30-year climatological time series (1986 to 2015).

### 3.4 Empirical estimation of lake volume, breach time, and peak discharge

We employ a set of empirical equations to estimate lake volume ($V$), breach time ($t_b$) and peak discharge ($Q_p$) in order to cross-check whether our simulation results are within the range of plausible values. Further, we use measured lake area ($A$) and
calculated lake volume ($V$) to estimate the mean lake depth ($D$). Following input data are derived from the analysis of pre- and post-GLOF satellite images and DEMs and used in the calculations:

- lake area before GLOF ($A$): 560,000 m$^2$
- lake width ($L_w$): 360 m
- breach depth ($B_d$): 20 m
- maximum breach width ($B_{w\_max}$): 80 m
- average breach width: 40 m (considering V-shaped cross-profile; $B_{w\_avg\_v}$), 60 m (considering trapezoidal cross-profile; $B_{w\_avg\_t}$)
- breach width at the base considering trapezoidal cross-profile ($B_{w\_base\_t}$): 40 m
- volume of material eroded from the dam ($V_{er}$): 300,000 m$^3$
- released volume ($V_{GLOF}$): 10,000,000 m$^3$

We employ a total of 14 different empirical equations to estimate $V$, 12 equations to estimate $t_b$ and 16 equations to estimate $Q_p$ (see Table 2 and Supplement Table 1). These equations are based on relationships derived from plotted values of well-documented cases. Despite the widths of uncertainty bands and prediction intervals of these datasets being high (Froehlich, 1995; Wahl, 2004), they represent useful tools for approximating the range of plausible values.



**Table 2. A list of empirical equations used to estimate V, $t_b$ and $Q_p$ (for detailed description of these equations see Supplement Table 1).**

| Characteristic | Used empirical equations |
|---|---|
| Lake volume: $V$ | Evans, 1986; O'Connor et al., 2001; Huggel et al., 2002; Wang et al., 2012; Fujita et al., 2013; Loriaux and Casassa, 2013; Emmer and Vilímek, 2014; Cook and Quincey, 2015; Kapitsa et al., 2017; Muñoz et al., 2020 |
| Breach time: $t_b$ | MacDonald and Langridge-Monopolis, 1984; Costa, 1985; Bureau of Reclamation, 1988; Von Thun and Gillette, 1990; Froehlich, 1995; Wahl, 2004 |
| Peak discharge: $Q_p$ | Kirkpatrick, 1977; Price et al., 1977; Soil Conservation Service, 1981; Bureau of Reclamation, 1982; Hagen, 1982; MacDonald and Langridge-Monopolis, 1984; Singh and Snorrason, 1984; Costa, 1985; Evans, 1986; Froehlich, 1995; Wahl, 2004 |

### 3.5 Process chain simulation with r.avaflow

We back-calculate part of the complex GLOF process chain, employing the simulation tool r.avaflow (Mergili et al., 2017; Pudasaini and Mergili, 2019; Mergili and Pudasaini, 2020). r.avaflow is a GIS-based open source simulation framework for multi-phase mass flows, which has the capacity to dynamically compute the interaction between landslides and lakes. Therefore, it is considered most suitable for the 2020 GLOF process chain at Jinwuco. In the present work, we back-calculate the release of the initial landslide into the lake, formation and spreading of displacement wave(s), dam breach, lake drainage and flood propagation 1.2 km downstream from Jinwuco. We do not consider the subsequent part of the process chain in the simulation continuing downstream, as quantitative reference data on travel times or discharges – and, farther downstream, also impact areas – are not available or at least highly uncertain.

Two scenarios are simulated, representing the two extremes in the range of likely triggers of the process chain (see Section 4.2 and Section 5.2 for more detailed explanations):

- Scenario A: The GLOF process chain is triggered by the release of the initial landslide into the lake. The full reconstructed landslide volume is considered, representing a very upper limit justified by the uncertainties in the volume calculation (Section 4.2).

- Scenario B: The initial landslide does not directly trigger the GLOF process chain and is therefore disregarded. Instead, the breach of the moraine dam is initiated by incision of the surficial outlet channel.

Apart from the initial landslide, other model inputs are the same for both scenarios: the terrain is represented by the reconstructed DEM (Sections 3.2 and 4.2), and so are the other input rasters: an initial landslide volume of 1.2 million m³ is released from the location shown in Fig. 6 in Scenario A, but not in Scenario B. A solid fraction of 80% with a density of 2,700 kg/m³ and a fluid fraction of 20% with a density of 1,000 kg/m³ are applied, assuming largely saturated conditions. According to topographic considerations and the volume range derived from empirical relationships (Sections 3.4 and 4.4), the volume of Jinwuco is set to 13.9 million m³ of water (fluid material with a density of 1,000 kg/m³). Erosion and entrainment of solid material from the terminal moraine is set to the extent of the calculated breach volume (300,000 m³, Section 4.2). For the erodible material, saturated conditions with 80% solid and 20% water are assumed. The same densities as for the landslide material are applied. The most important model parameters are the internal and basal friction angles of the solid material $\varphi$ and $\delta$, the Manning number $n$, and the coefficient of erosion $C_E$. Empirically adequate results are obtained with $\varphi = 25°$, $\delta = 10°$, $n = 0.05$, and $C_E = 10^{-4.25}$. This combination of parameters is probably not the only one possible due to equifinality (Beven, 1996). Nevertheless, it is considered physically plausible, and a detailed sensitivity analysis would go beyond the scope of the present work. The process duration considered in the simulation is $t = 7,200$ s (two hours) from the release of the initial landslide. All simulations are run at a raster cell size of 12 m.





**3.6 Analysis of impacts on society**

On-site photos, videos, and local news reports are collected to capture and quantify the immediate societal impacts of this GLOF event. Access to this information is largely made possible by the rapid development of online personal sharing applications (e.g. Douyin or Tiktok) in recent years. Here, we gather an abundance of photos and videos from locals and travellers that can provide the most intuitive in situ information about the lake and the region.

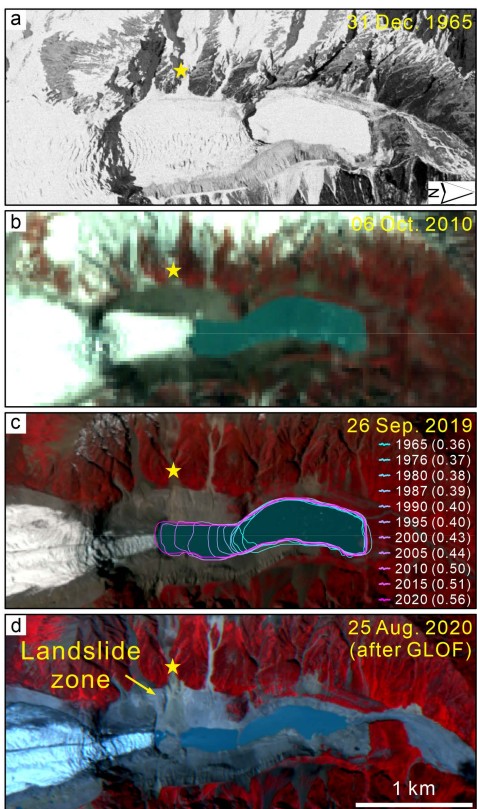

**Figure 3. Satellite images documenting the evolution of Jinwuco during 1965–2020 and the location of the debris landslide. The lake area in parentheses of panel (c) is given in square kilometers. The backgrounds are from Corona KH-4A, Landsat-5 TM, Sentinel-2A and Ziyuan-3, respectively.**

**4 Results**

**4.1 Long-term evolution of Jinwuco and adjacent slopes**

Historical satellite images reveal that Jinwuco has undergone significant areal expansion since 1965 (Fig. 3). In 1965, its surface area was $0.36 \pm 0.003$ km$^2$. By 10 May 2020, the lake area has increased to $0.56 \pm 0.02$ km$^2$, showing an increase of approx. 56% relative to 1965 (Fig. 4). Overall, the surface area of Jinwuco shows a non-linear growth trend from 1965 to 2020

– with an accelerated growth rate observed from 1995. Importantly, the rear part of the lake, where the landslide deposited, is shown to have developed since 2010 (Fig. 3). After the GLOF event on June 26, 2020, the lake area is reduced by approx. 54% to $0.26 \pm 0.01$ km$^2$ (Fig. 4).





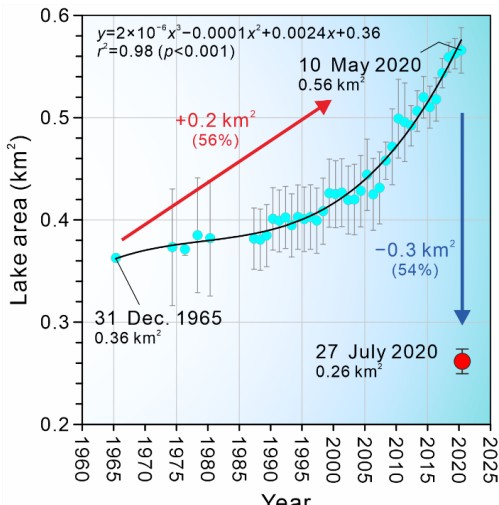

**Figure 4. Inter-annual dynamics of Jinwuco during 1965–2020 (prior to the GLOF), and the abrupt decrease in lake area after the GLOF.**

**4.2 Reconstruction and quantification of relevant features**

The reconstruction of the main features allows for drawing some conclusions about the mechanisms involved in the GLOF process chain, and to estimate the mobilised volumes (Fig. 5 and Fig. 6):

1. Lake: the pre-event lake area was approx. 560,000 m² (Fig. 4), with a maximum width of ca. 360 m. A lake volume of 13.9 million m³ is reconstructed (Section 4.4), corresponding to a maximum lake depth of 54 m. The pre-event lake level is set to 4,444 m, whereas the post-event lake level is set to 4,424 m, resulting in a drop of 20 m during the event. In the frontal portion of Jinwuco, some discrepancies exist between the post-event lake extent derived from the DEM and the satellite image (Fig. 5): this mismatch most likely indicates inaccuracies of the DEM.

2. Landslide: a landslide release volume of approx. 1.2 million m³ is estimated from the DoD. However, there is an error margin of few 100,000s of m³ due to the non-perfect overlay of the two DEMs. Despite some uncertainty in relation to the exact volumes involved, it is quite clear from the DoD that the landslide initiated at the western lateral moraine slope in the very rear part of Jinwuco. The satellite image reveals that there might have been some indirect involvement of material from slopes located above, which accumulated behind the crest of the lateral moraine (possibly even slightly overtopping it). The additional load, in connection with saturation of the lateral moraine, might have triggered the landslide. However, this interpretation remains speculative. The landslide deposit is clearly visible in the rear portion of the post-event lake in the satellite image, and is also clearly represented in the post-event DEM. It seems likely that the landslide did not deposit onto the glacier tongue and therefore entered directly into the lake (Fig. 5). A deposition volume of 1.0 million m³ is derived. This number is uncertain not only due to possible errors of the DoD, but also as part of the deposit is covered by lake water, and part of it might have been washed away. However, the order of magnitude of the deposition volume corresponds very well to the release volume, so that the general pattern appears fairly robust.

3. Breach of the terminal moraine: a maximum breach depth of 25 m and an eroded volume of 300,000 m³ are derived from the DoD. The difference of 5 m between the maximum breach depth and the drop of the lake level is a result of the steeper longitudinal profile of the upper part of the drainage channel after the breach, compared to the situation



before the breach. However, as it is clearly shown in Fig. 6, there is a vertical offset between the pre- and post-event DEMs, limiting the accuracy of the estimates. Even though the offset is not constant in space, it is likely that the pre-event DEM would have to be uplifted in the area of the terminal moraine, relative to the post-event-DEM, to bring

the two in line, which would increase the maximum breach depth by up to approx. 7 m. However, the data basis is too insufficient to justify such a correction.

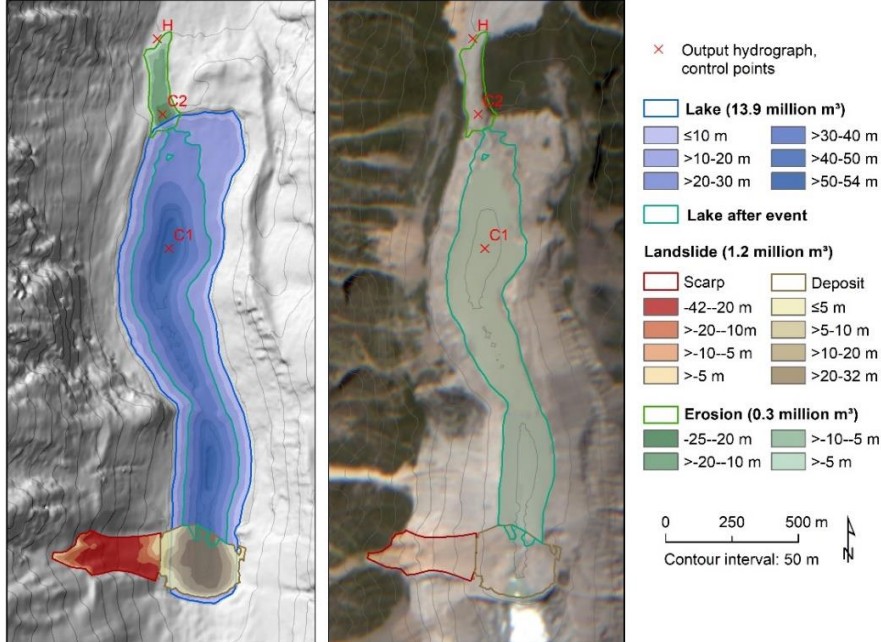

**Figure 5. Reconstruction of the lake depth, landslide, and breach of the moraine dam. Base map is shaded relief map (left) based on the DEM generated from Ziyuan-3 stereoscopic pairs, and the Ziyuan-3 satellite scene (right) acquired**
**on 25 August 2020.**

Analysis of the Sentinel-1 radar imagery reveals that landslide activity impacting the lake has started already in the period between 9 June and 21 June 2020 (Fig. 7). The pattern in the rear part of the lake visible in the scene of 21 June most likely represents a landside deposit, possibly of significant size, even though the available data do not allow for volume quantification

at this point. However, a scenario whereby the GLOF process chain was maybe not directly triggered by the landslide appears likely (Section 3.5 and Section 5.1).

(a) Erosion depth at dam derived from elevation data

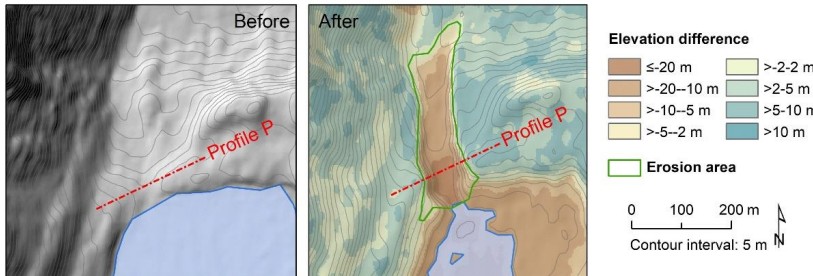

(b) Cross section of erosion channel at profile P

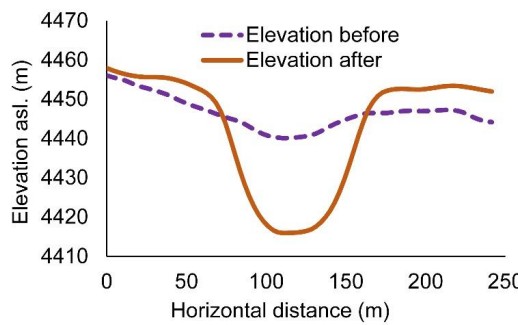

**Figure 6. Breach of the moraine dam of Jinwuco. (a) Situation before and after the event of 26 June 2020, with difference in elevation between the DEMs derived before and after the event. (b) Cross section along the profile P (see (a)) before and after the event. Topographic base map is based on the DEM generated from Ziyuan-3 stereoscopic pairs.**

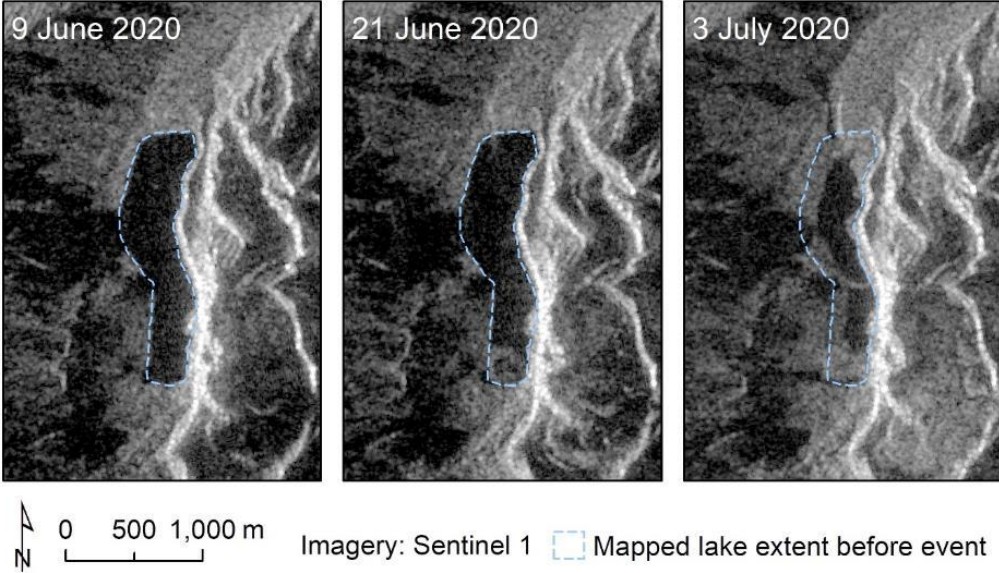

**Figure 7. Sentinel-1 radar satellite images of Jinwuco and surroundings representing the situation on 9 June, 21 June, and 3 July 2020.**





### 4.3 Meteorological conditions before the event

When compared to the longer-term climatology it becomes immediately obvious that the landslide and GLOF process chain

at Jinwuco occurred during a period of unusually warm and wet weather (Fig. 8). Over the dry winter and into the pre-monsoon months, precipitation measured at Lhari station was consistent with the longer-term mean, as seen in the accumulated totals (Fig. 8b). However, through May and particularly June 2020, notable periods of extremely heavy precipitation occurred, regularly exceeding 90th and even 95th percentile levels. In June, the 1986–2015 average precipitation total was 141 mm, compared to 240 mm in 2020, corresponding to an anomaly of +70%. A particularly wet period is observed from 5–8 June

2020, with a total of 88 mm and a daily maximum of 31 mm of precipitation on 7 June 2020. This wet period directly preceded the time window in which the landslide into the lake is identified in Fig. 7 (Section 4.2). Temperatures transitioned from unusually cold conditions in late May and early June (preceding the landslide window), to anonymously warm temperatures from mid-June onwards, peaking at an extremely warm daily maximum of 19.4°C on 24 June, immediately prior to the GLOF event. During the cold period prior to the landslide window, minimum temperatures were at or below freezing, suggesting

some of the heavy precipitation from 5-8 June would have fallen as snow around Jinwuco. Subsequent gradual snowmelt could provide a plausible explanation for the lag between this heavy precipitation episode, landslide initiation, and eventual moraine breach, given that the GLOF event was preceded by two relatively dry, yet warm days (Fig. 8a).

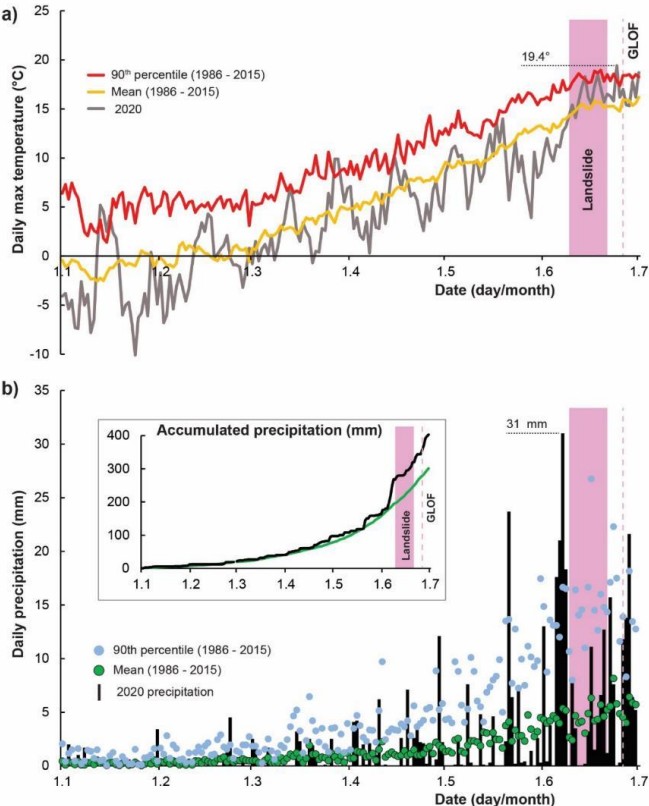

**Figure 8. Meteorological conditions from January - June 2020, measured at the Lhari weather station, compared to the longer-term climatology (1986–2015).**

### 4.4 Empirical estimation of lake volume, breach time, and peak discharge

Out of 15 lake volume equations, 13 yielded meaningful results, ranging from 8.3 to 26.6 million m³. Mean calculated lake

volume is 15.0 million m³ and the median value is 14.1 million m³. This compares favorably to a reconstructed volume of 13.9
million m³ used in the process chain simulations (Section 3.6). The first (Q1) and the third quartile (Q3) are 13.2 million m³
and 15.2 million m³ respectively (Fig. 9a). Accordingly, calculated mean depth of the lake ranges from 14.8 to 47.6 m (average
mean depth 26.7 m, median mean depth 25.2 m, Q1 is 23.6 m and Q3 is 27.2 m). Calculated breach times range from 0.28 to
1.76 hours, with 11 out of 12 calculated values suggesting a breach time <1 hour, with the mean value of 0.62 hours and a

median of 0.47 hours (0.61 hours and 0.43 hours, respectively, if equations for erosion resistant dam are excluded; Fig. 9c, d).
Calculated peak discharge at the dam ranges from 2,354 m³/s to 11,832 m³/s. The average value of calculated peak discharges
is 5,390 m³/s while the median is 4,406 m³/s. Q1 and Q3 are 3,029 m³/s and 6,662 m³/s respectively (Fig. 9b).

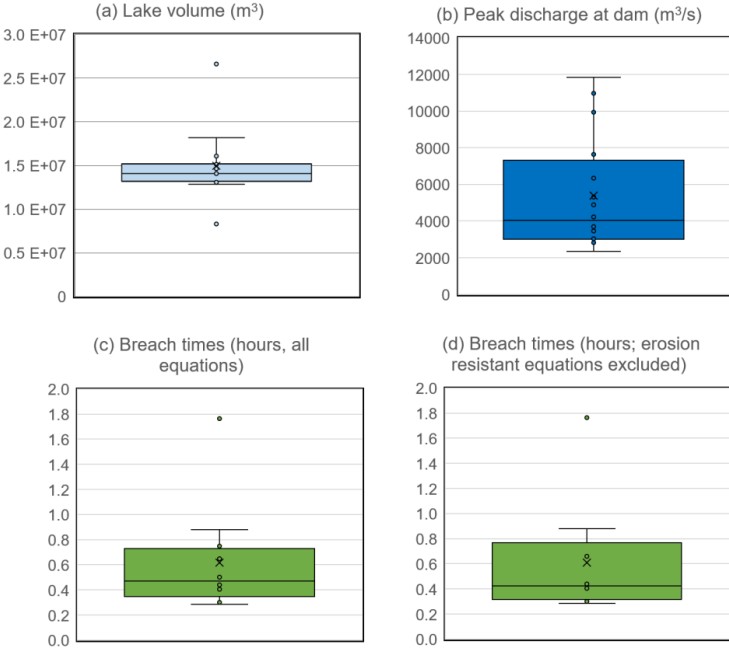

**Figure 9. Lake, dam breach and flood parameters, derived from sets of published empirical relationships. (a) Volume of Jinwuco. (b) Peak discharge at the dam. (c) Breach time, considering all available equations. (d) Breach time, considering only those equations derived for easily erodible dams (an assumption we consider likely to hold for the dam of Jinwuco).**

### 4.5 Process chain simulation with r.avaflow

A detailed look at the dam breach (Fig. 10) reveals that the simulation results for both scenarios A and B are in good
correspondence with the ranges of peak discharge and breach depth derived through the empirical relationships described in
Section 4.4 and illustrated in Fig. 9. For Scenario A: at the point C2 (Fig. 5 or Fig. 11), depth erosion is completed at
$t = 23$ minutes after landslide release, i.e. 21–22 minutes (0.36 hours) after the impact wave reached the outlet of the lake,
initiating retrogressive erosion. This time span is well within, even though in the lower part, of the range revealed in Fig. 9d. After the breach has developed to its full depth, lake drainage continues and asymptotically approaches the level of the eroded gully, which is almost reached after two hours (Fig. 10a). The peak discharge associated with the landslide-induced impact wave reaches 10,900 m³/s after 100 s (computed at a higher temporal resolution, therefore not visible in Fig. 10b). Afterwards, discharge decreases and then increases again due to retrogressive erosion of the moraine dam. The simulated peak discharge during the retrogressive erosion stage reaches 5,000 m³/s at $t = 24$ minutes, being well within the range indicated in Fig. 9b. With dropping lake level, the simulated discharge slowly decreases to a small residual value after two hours (Fig. 10b). At all stages, the simulated flow is dominated by water, with a peak of approx. 240 m³/s of solid discharge at $t = 19$ minutes, when retrogressive erosion is the most intense.

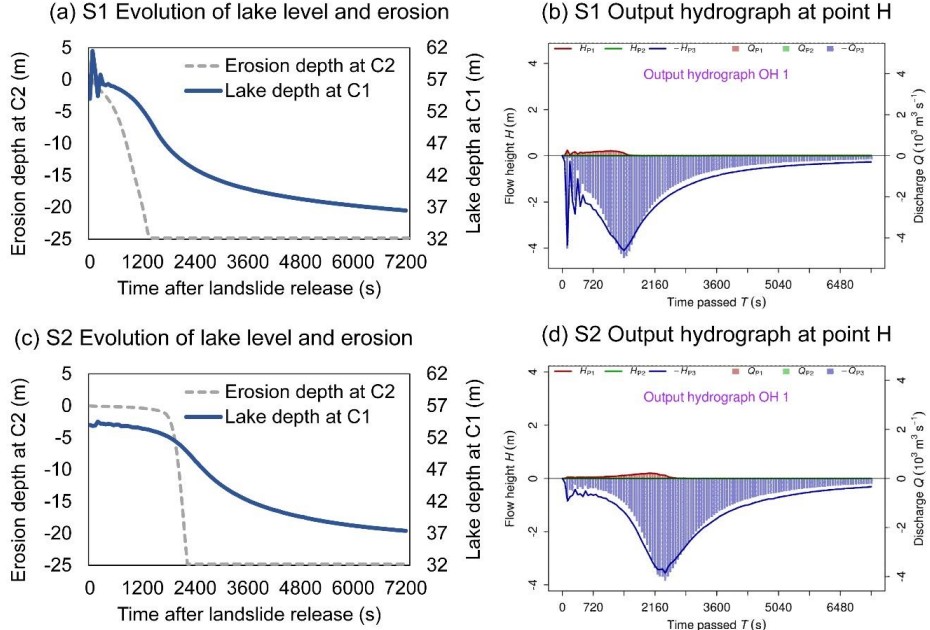

**Figure 10. Dam breach as simulated with r.avaflow. (a) Scenario 1, evolution of lake level and the erosion channel. (b) Scenario 1, output hydrograph. (c) Scenario 2, evolution of lake level and the erosion channel. (d) Scenario 2, output hydrograph. Locations of the points C1, C2, and H are indicated in Figure 5 and Figure 11.**

The patterns yielded with the simulation of Scenario B (Fig. 9c and d) are similar in general, with the exception of the initial impact wave that is only observed in Scenario A. However, onset of erosion takes more time, so that the breach and associated emptying of the lake are delayed. The tipping point where slight channel erosion turns into a rapid dam breach occurs at approx. 30 minutes after the start of the simulation. Thereafter, the base level of the erodible material is reached at $t = 37$ minutes (0.62 hours – within the range indicated in Fig. 9d). The peak discharge of 4,200 m³/s is reached at $t = 40$ minutes. As is also the case in Scenario A, the flow is entirely dominated by fluid: solid discharge peaks at 210 m³/s at $t = 36$ minutes. Emptying of the lake continues at a decreasing rate afterwards. The lake level has dropped by 18 m after two hours, meaning that the final drop of 20 m has not yet been reached.



Fig. 11 illustrates the evolution of the GLOF process chain for Scenario A (Section 3.6). This simulation only considers the section until the floodplain downstream of the moraine dam of Jinwuco. Initially, the landslide impacts the lake ($t = 10$ s),

inducing a push wave that runs up the opposite slope ($t = 20$ s). The minor outflow at the moraine dam represents the usual surficial lake drainage and is not related to the landslide impact. The impact area shown at $t = 60$ s and the increased lake width near point C1 reveal the evolution of the impact wave towards the front of Jinwuco. Still, at this point there is no major signal of the impact wave at the outlet of the lake. The maximum height of the impact wave at C1, with 8.5 m, is reached at $t = 70$ s. At $t = 85$ s, a sudden increase of flow height at the point C2 from 0.4 m to 9.7 m within 10 s indicates the onset of overtopping.

Retrogressive erosion takes place along the main outflow channel, induced by the signal of the initial impact wave. Some of the wave energy is reflected at the shore of the lake towards the terminal moraine, so that a secondary wave with a height of 5.4 m passes through C1 at $t = 125$ s. Linear retrogressive erosion of the moraine dam leads to a continuous lowering of the base of the outflow channel, and consequently, lowering of the lake level. Two hours after the onset of the initial landslide, the lake has shrunk to both a size and level which come close to the post-event observations. Until this point of time,

approx. 9.2 million m³ of water have drained through the hydrograph profile at point H (Fig. 10), according to the simulation. The simulated impact area on the floodplain immediately downstream of the dam corresponds fairly well to the observed impact area in that section.

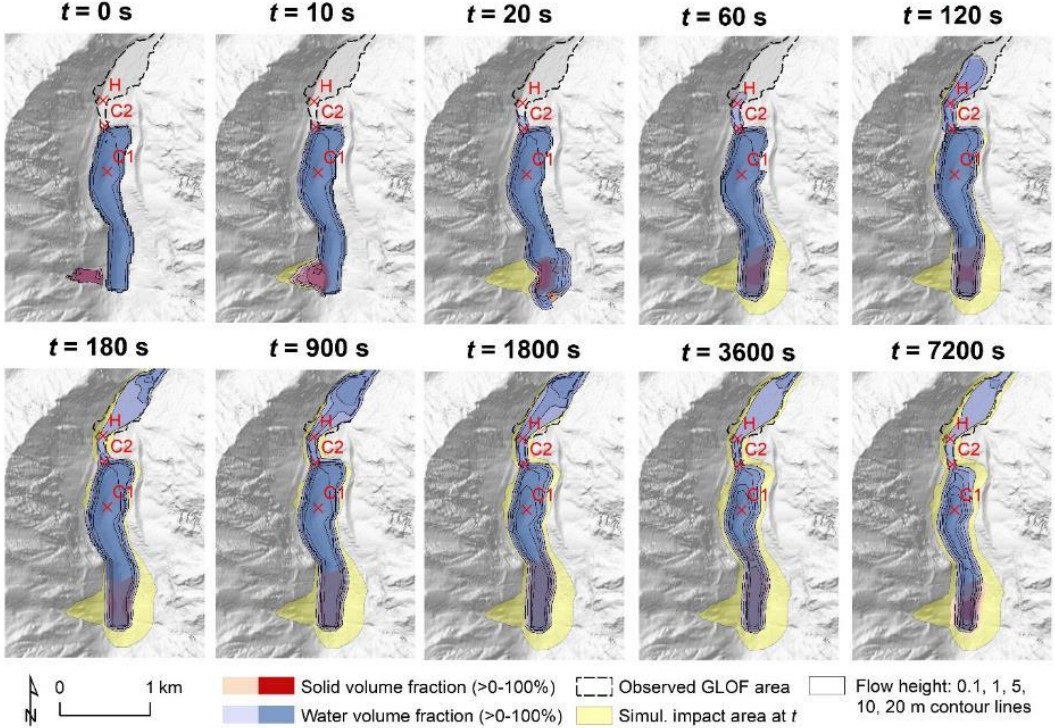

**Figure 11. Scenario A: evolution of flow height and GLOF impact area as well as solid and water volume fractions during the first two hours of the event, as simulated with r.avaflow. Only the section until the floodplain immediately downstream from the moraine dam is considered in this simulation. Base map is shaded relief map based on the DEM generated from Ziyuan-3 stereoscopic pairs.**





**4.6 Analysis of impacts on society**

According to the news report from the Lhari Radio and Television Station on 15 July 2020, this GLOF event caused severe destruction to villages and infrastructure to various degrees on both sides of the downstream river. The destruction consists mainly of damage to buildings, roads, bridges, or farmlands, just to name a few (detailed statistics are provided in Table 3). Fortunately, this event did not lead to any fatalities or injuries largely because it took place during the daytime, and the villagers

collecting medicinal herbs upstream observed the flood and promptly informed the downstream villages to evacuate. The affected area includes almost all the villages on both sides of the river, especially Baixiongcun and Zibucun. These where are also the most affected by the 2013 Ranzeriaco GLOF (Fig. 1; Liu, 2014). The power of the flood is noticeably weakened after its confluence with the Gagring Qu in the Zhongyucun and hence does not result in much damage to areas further downstream. This flood damaged a major proportion of 43.9 km of roads between Yigacun and Zhongyu township (Fig. 1; Fig 12), as well

as the only steel bridge to the township office (Fig. 13a), which caused certain difficulties for timely emergency rescue operations.

**Table 3. Details of the damages and losses caused by the GLOF of 26 June 2020.**

| Item | Description[1] |
|---|---|
| Casualties | No casualties caused in this GLOF. |
| Buildings | 10 residential houses destroyed and 1 damaged. |
| Roads | 43.9 km of road connecting from Zhongyu township government seat to Yigacun was almost washed away. |
| Bridges | 6 steel bridges, 1 suspension bridge and 1 concrete covered culvert were washed away. |
| Farmlands | Flooded farmland 19.98 hectare (including 2.58 hectares of rapeseed fields); destroyed farmland 6.24 hectare (including 0.55 hectares of rapeseed fields); destroyed farmland fence 8,517 m. |
| Grasslands and forage lands | Destroyed artificial grassland area 3.15 hectare; flooded forage land 2.58 hectare. |
| Forests | Destroyed 9.07 hectare of forests for soil and water conservation project. |
| Flood control levee | Breach of 7.7 km of flood control levee. |
| Others | The total CNY 8.4 million 45% completed Yiga view project is completely flooded; 25 corrugated pipes, 3 culvert access bridges, and 7 simple bridges for humans and livestock leading to summer pasture were washed out. |

[1]Data from Lhari County Radio and Television Station.


Fig. 12 shows the valley and river channel below Jinwuco before, during and after the GLOF. It is evident from the comparison of the photographs that the floodwaters almost inundated the wide river valley downstream and reached the location of Yigacun, located in a relatively high position. The large elevation difference between the surface of Jinwuco and the downstream river channel (approx. 630 m) provides significant potential energy, setting the stage for enhanced erosion and greater damage.

Further, the flood caused severe erosion of the lower valley and alteration of the river channel below through the formation of alluvial fans. Fig. 13 gives a picture of the situation during the flood further downstream at the location of Zhongyucun. The aerial photograph shows that the massive impact of the flood washed away roads, bridges and put some buildings along the river at risk. The Zhongyu township has experienced rapid development and expansion of infrastructure over the past fifteen years (some examples are shown in Fig. 14). Such development has increased the level of exposure to potential GLOFs, as

shown in Fig. 13c: these buildings were constructed recently in areas located directly on the riverbanks and exposed to flooding.





**Figure 12.** The situations in the valley and river channel below Jinwuco before, during, and after the GLOF. (a) Photo taken from Yiga Glacier looking to Yigacun. Insets showing the situations in the valley during and after the GLOF (photos: Q. Quying and Y. Quzhen). (b) The river channel and road before the GLOF taken from Yigacun looking upstream. The yellow dashed line is the flood flow direction. (c) A view taken from upstream looking to Yigacun. Insets showing the situations in the river channel during the GLOF (photos: Q. Quying). Aerial photos from 2 May 2020 were taken by K. Liang.




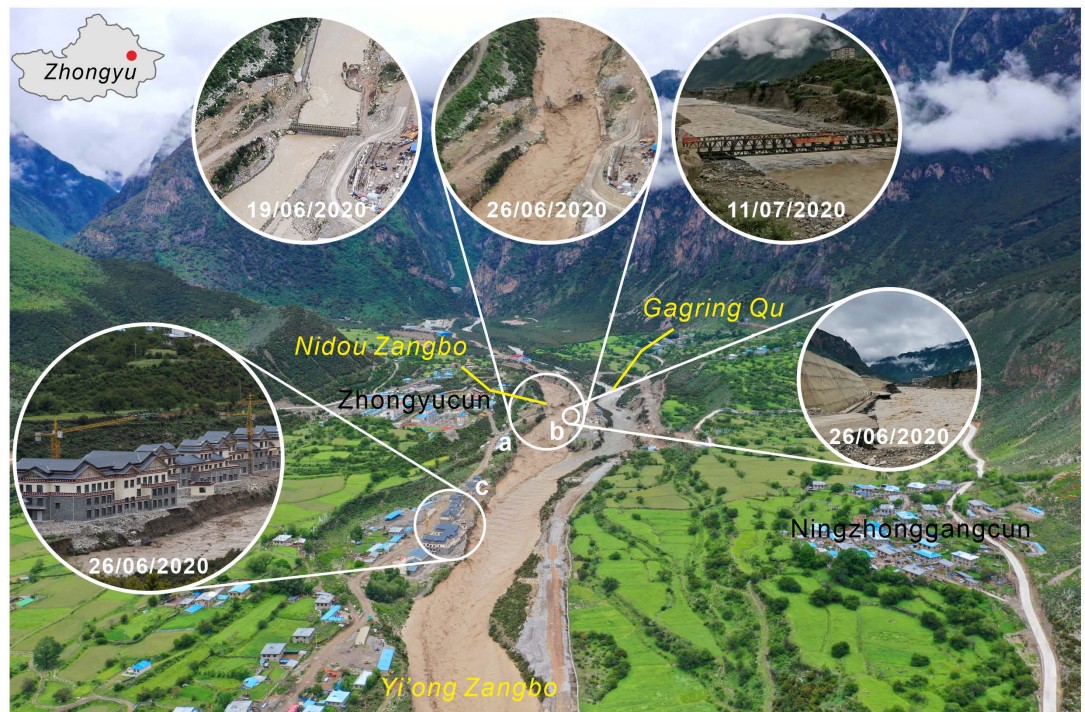

**Figure 13. Examples of GLOF damage to downstream infrastructure and buildings at the confluence of Nidou Zangbo and Gagring Qu. Background aerial photo by H. Li taken on 26 June 2020. (a) Comparison of the bridge connecting Zhongyucun and Ningzhonggangcun before, during (photos: H. Li) and after the GLOF (rebuilt, photo: K. Ma). (b) Destroyed roads (photo: H. Li). (c) Affected buildings at risk (photo: H. Li).**

## 5 Discussion

### 5.1 Reconstruction and timing of the event

Quantification of landslide and erosion extent and volumes are to be interpreted as rough estimates rather than exact values, given the hard-to-quantify uncertainties in the available pre- and post-event DEMs. Equally, the mechanism connecting the landslide and the moraine breach remains the subject of discussion. There is clear evidence of material having entered the rear section of the lake days before the recorded GLOF, leaving no signs of associated drainage and erosion (Fig. 7). Part of the total landslide material had mobilized and entered the lake between 9 June and 21 June 2020. Most likely, this process happened slowly, potentially enhanced by ongoing smaller rainfall events and snowmelt, probably causing increased outflow but no impact wave or notable erosion. The scenes of 21 June and 3 July 2020 strongly reveal that a major part of the landslide deposit could already have been present in the lake on 21 June. This assumption is supported by the comparison of the landslide deposit derived from the post-event DEM and the pre-event lake level. However, there is no means to quantify the volume deposited by 21 June. Available data reveal three ways of interpreting the sequence of events:

- An additional, more rapid mobilization of a significant volume of debris entered the lake on 26 June, triggering an impact wave and the subsequent processes.
- There was no linkage between the landslide and the GLOF. Breach of the moraine dam was instead initiated by another trigger, such as increased outflow and/or weakening of the dam due to excessive rainfall and snowmelt.





•    The first hypothesis is not strongly supported by the data, whereas the temporal coincidence of landslide and GLOF, with a time lag of only 5–17 days, makes the second hypothesis also not very convincing. Hence, a combination of both factors might be a third possibility: pre-weakening of the dam due to landslide-induced increased outflow could have facilitated the onset of erosion by subsequent rainfall, snowmelt and/or by a smaller secondary landslide or calving event into the lake.


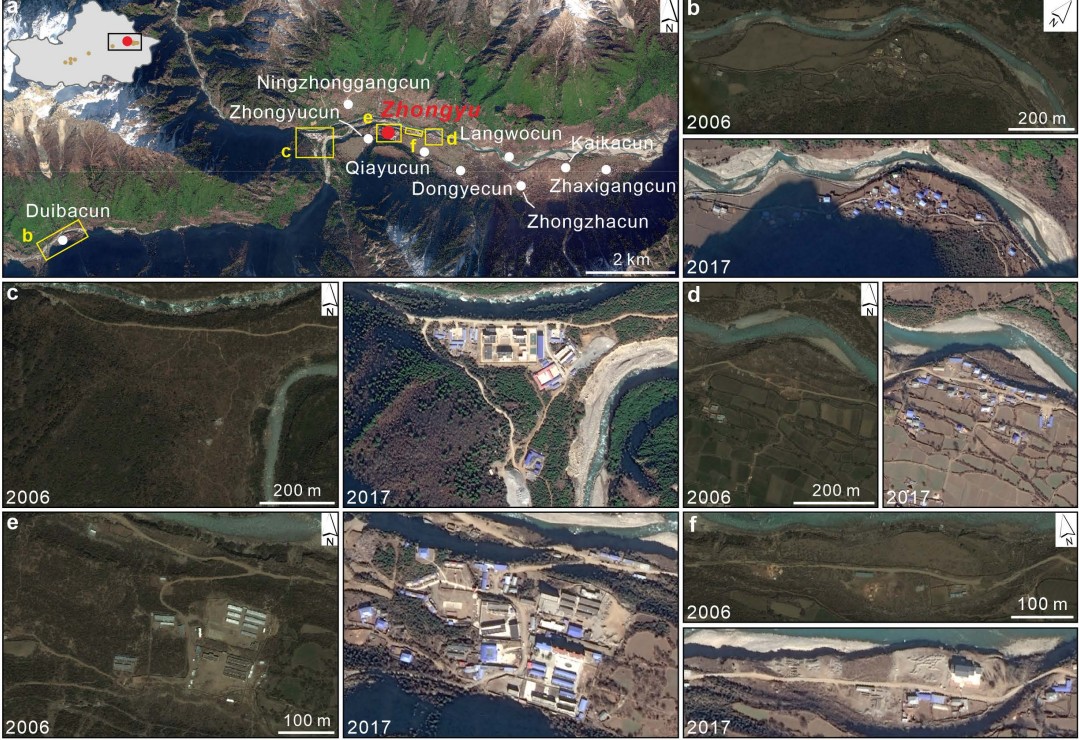

**Figure 14. Examples of rapid development and expansion of the villages of lower Zhongyu township during 2006–2017. The background of panel a and the satellite scenes for 2017 panels are a true-colour composite Gaofen-2 image acquired on 7 December 2017. The satellite images for the 2006 panels are from © Google Earth acquired on 30 April 2006.**


While the available evidence does not permit to exclude any single hypothesis, the meteorological conditions in June 2020 (Fig. 8) strongly supports the third hypothesis. We consider it likely that the extremely high precipitation in the period from 5–8 June has triggered slope instability in the western lateral moraine area. However, this has not resulted in immediate rapid failure, but rather delayed, gradual landslide processes, as supported by the evidence. Further rainfall and snowmelt under

unusually warm conditions after 8 June may have enhanced this gradual destabilisation of the moraine. The rather dry and extremely warm conditions immediately prior to the GLOF may have resulted in increased melting and thawing processes, possibly triggering new instabilities (including calving) into the lake and/or an increased lake level, leading to the final tipping point of a system which had already been at the verge of collapse.

The impossibility of determining the cause of a GLOF is not new to science: the most prominent previous case was the 1941

event at Lake Palcacocha in the Peruvian Andes (Mergili et al., 2020), where there was speculation about triggering of the





catastrophic moraine dam breach by calving or an ice avalanche, however poorly supported by evidence. The reverse issue is encountered with the event at Jinwuco: there is clear evidence of a potential GLOF-triggering landslide, however with a possible time lag between the landslide and the GLOF that can be plausibly explained in the context of meteorological drivers. Comparable cases have not yet been documented, so that more research will be necessary to shed light on this type of

phenomenon.

**5.2 Empirical estimations and process chain simulation**

An extensive literature review with regard to the empirical estimation of lake volume, breach time, and peak discharge resulted in a broad range of possible values (Fig. 9). Both of the simulation scenarios A and B have provided plausible characteristics of peak discharge and breach time with the chosen parameter combinations. Due to the strong signal of the impact wave at the

dam in Scenario A, there is a more rapid onset and progress of erosion, so that the peak discharge is higher and the breach time shorter, compared to Scenario B. However, the general pattern is the same. Model setup has revealed two critical aspects for the simulation:

- The results are very sensitive to the empirical entrainment coefficient, which is multiplied with the flow momentum in order to determine the erosion rate. In general, scaling of the erosion rate with flow momentum, though simple and

straightforward, might not be the mechanically most appropriate way and could be replaced in the future by an excess shear stress approach, as it is used in some single-phase models (e.g. Iverson and Ouyang, 2015).

- In the present work, erosion and entrainment is restricted to the breach reconstructed through the overlay of the pre-event and post-event DEMs. This constraint is introduced as the internal structure of the moraine dam is unknown, and it is assumed that the part most susceptible to erosion was actually eroded during the event on 26 June 2020.

However, such simplification limits the possibility to evaluate the performance of the entrainment model. Strategies for a more detailed parameterization of dam erodibility are still the subject of future research, and would also be important to make predictive simulations of dam breach more reliable.

**5.3 The potential role of anthropogenic climate change on the process chain**

The Jinwuco process chain described herein provides a strong case for attributing the role of anthropogenic climate change as

a direct contributor to a GLOF-related disaster. Here, we can consider the lake expansion not only as a driver of potential outburst magnitude and downstream impacts (Huggel et al., 2020), but as a direct enabler of the process chain. Simply stated, prior to 2010, the process chain as described here could not have initiated – the landslide would have ran harmlessly across the glacier tongue with minimal (if any) debris depositing into the lake. The nearest weather station at Lhari County has recorded a significant warming rate of 0.5°C per decade over the last 40 years (Fig. 1), reflecting a general increasing warming

trend seen across Tibet, with the most rapid warming observed since 2000 (You et al., 2016). These warming trends correspond with widespread glacial retreat and lake expansion across Tibet (Bolch et al., 2019), and particularly align to the period of accelerated expansion of Jinwuco (Fig. 4). While regional studies are lacking, the strong anthropogenic signal seen in warming over Asia (Bindoff et al., 2013), coupled with the increasing contribution of anthropogenic forcing attributed to global glacier mass loss over the past century (Marzeion et al., 2014), gives us a high level of confidence that the accelerated expansion of

Jinwuco has at least partially been driven by anthropogenic greenhouse gas emissions. Beyond the direct role in allowing the debris landslide to strike the expanding lake body, the retreat of the Jinwu Glacier has likely also played a primary role in destabilising the lateral moraine from which the landslide initiated (McColl, 2012; Klimeš et al., 2016). On the other hand, although extremely heavy precipitation appears to have played a major role in triggering the landslide event, trends of such extremes over Tibet are generally mixed or not statistically significant (Zhan et al., 2017).



### 5.4 Implications for future GLOF hazard and risk assessment

Steep lateral moraine walls are commonly observed adjacent to receding glacier tongues and proglacial lakes globally, and on a regional scale, lessons learned from the Peruvian Cordillera Blanca suggest that landslides from moraines are the most frequent GLOF trigger in later stages of glacier ice loss (Emmer et al. 2020a). Yet surprisingly, instabilities from lateral moraines have rarely been associated with large outburst disasters (Klimeš et al., 2016). Existing GLOF hazard assessment schemes (e.g. GAPHAZ, 2017) and related modelling studies (Schneider et al., 2014; Schaub et al., 2016), have typically emphasised the importance of rapid process chains relating to large rock and ice instabilities that enter a lake at high velocities generating massive flood waves. However, we would argue that results from this study should draw renewed attention to the possibility of complex, lagged process chains involving large moraine wall instabilities, noting that the initial impact and dissipation of the wave energy may not be the end of the story. Such process chains may become of increasing importance in the future, given further glacial retreat or downwasting (Emmer et al., 2020b), leading to higher moraine walls being precariously positioned above expanding proglacial lakes. Furthermore, under a future climate, triggering of moraine instabilities under changing conditions could become more likely (Panday et al., 2015; Sanjay et al., 2017) .

Beyond the hazardous process chain, downstream exposure of buildings and other infrastructure has been shown once again to be a key driver of GLOF risk (Kala, 2014; Huggel et al., 2020). Hence, there is an urgent need to ensure lives and livelihoods are better protected through effective hazard mapping and land use planning, coupled with comprehensive early warning and response strategies to minimise GLOF risk even under worst-case future climate scenarios. In the specific case of Jinwuco, although the significant incision of the moraine dam now limits the maximum depth of the lake, the lake area and volume will increase further as the glacier retreats, meaning the potential future outburst risk will continue to evolve. Furthermore, considering simple diagnostic topographical criteria (after Frey et al., 2010),  there is a strong likelihood of a new glacial lake developing on the flat plateau higher on the Jinwu Glacier in response to a future retreat, meaning that anticipatory forward-looking approaches to risk management are required.

### 6 Conclusions

We have comprehensively analyzed the Jinwuco GLOF process chain that occurred on 26 June 2020 in eastern Nyainqentanglha, Tibet, China, based on remotely sensed data, eye-witness accounts, news reports, and computer simulations. The main findings of this work can be summarized as follows:

1. While most historical moraine-dammed GLOFs in the region have been typically triggered by an instantaneous ice/rock avalanche into a lake, in the case of Jinwuco, an initial slope failure from a steep lateral moraine was considered part of a gradual process chain: analysis of meteorological conditions from some weeks before the GLOF suggests that, most likely, the extremely heavy rainfall and high air temperatures in June were essential drivers of this process chain. Analysis of Sentinel-1 images revealed a time lag of 5-17 days between the rainfall-triggered landslide and the GLOF, the latter coinciding with warm and dry conditions. We therefore conclude that effects of subsequent rainfall, snow melt, a smaller secondary landslide, or calving led to a tipping point where retrogressive erosion of the moraine dam, possibly pre-weakened due to the effects of the landslide, was induced.

2. Two scenarios considered in the back-calculation of the outburst process chain produce plausible results that are consistent with empirically derived ranges of peak discharge and breach time. However, given the likely chain of events derived from the Sentinel-1 images, and meteorological analyses, we consider Scenario B (retrogressive erosion of the dam without a major impact wave) to be a more realistic possibility. This scenario predicts a peak discharge of 4,200 m³/s, reached at 0.67 hours at the dam.





3. The long-term evolution of Jinwuco and its relationship to the likely landslide trigger, as revealed by a time series of
satellite scenes, provide strong evidence that this event is directly related to anthropogenic climate change. Climate
warming has led to the continued retreat of glaciers like Jinwu in the region, while associated lakes have expanded
and the stability of the surrounding slopes has been altered. Given the fact that instabilities from the flanks of a lake
may increase under a warmer and wetter future climate, the Jinwuco case provides a timely reminder of the need for
comprehensive and forward-looking hazard and risk assessment frameworks, considering complex instantaneous and
gradual process chains.

*Data availability.* All satellite images and weather station data used in this study can be obtained free of charge or with
reasonable request. Additional datasets used are available from the corresponding authors on request.

*Code availability.* The open source mass flow simulation tool - r.avaflow, and related manual, training data and the necessary
start scripts can be obtained from Mergili and Pudasaini (2020, https://www.avaflow.org/).

*Author contributions.* G.Z. and S.A. conceived the ideas and co-designed the research with M.M. G.Z. collected and prepared
the input data and performed analysis of lake evolution and societal impacts of the flood. M.M. provided the simulation tools
and made the simulation of the outburst process chain. A.E. quantified flood-related parameters. S.A. carried out the
meteorological analysis with discussions and contributions from G.Z., M.M. and H.G. A.B. provided financial support and
supervised this work with M.S. G.Z., M.M., A.E. and S.A. wrote the manuscript and prepared the charts. All authors discussed
the results and commented on the manuscript.

*Acknowledgements.* We appreciate the U.S. Geological Survey (USGS), European Space Agency (ESA), and China Center for
Resources Satellite Date and Application (CRESDA) for allowing free access to their satellite archives. Special thanks to the
China Remote Sensing Satellite Ground Station (RSGS) – Kashi Ground Station, who provided Ziyuan-3 satellite data after
the flood and two DEMs before and after the flood. Special thanks also to Q. Quying, Y. Quzhen, K. Liang, H. Li and K. Ma,
who provided valuable field materials and flood-related information. This work was funded by the Strategic Priority Research
Program of Chinese Academy of Sciences (Grant No. XDA20030101). The contribution of S.A. was partially supported by
the Swiss National Science Foundation (Grant No. IZLCZ2_169979/1), and counterpart grant of the National Natural Science
Foundation of China (Grant No. 21661132003). A special acknowledgement to China-Pakistan Joint Research Center on Earth
Sciences that supported the implementation of this study.

*Competing interests.* The authors declare that they have no conflict of interest.



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
