# Peer review of "The 2020 glacial lake outburst flood at Jinwuco, Tibet: causes, impacts, and implications for hazard and risk assessment"

_The Cryosphere, 2020_

## Author Comment (AC2)

21st April 2021

**Sub: Response to referees' comments for manuscript TC-2020-379**

**General response**

Dear referees,

We are very grateful to you for your positive, very constructive and detailed reviews on our manuscript. We have carefully studied these reviews and have addressed or clarified all of them. Our point-by-point responses are attached below in blue text (revisions in red), while the original referees' comments and manuscript text are in black.

Overall, we fully agree with most of the comments raised by you (as detailed below) and have revised our manuscript accordingly. In summary, our revisions have mainly focussed on two points:

(1) The lag time between the landslide and the GLOF: Through the analysis of more satellite images, we realized that our previous interpretation of the radar images needs to be reconsidered because an additional optical image acquired on 24 June 2020 showed that the landslide had not yet occurred at this time. This means that the landslide could have occurred either on the same day as the GLOF or at most two days before it, suggesting a significantly reduced lag time and closer relationship between these two events. Given the available evidence, however, we believe that both scenarios considered (Scenario A: landslide-induced impact wave with overtopping and resulting retrogressive erosion of the moraine dam; Scenario B: retrogressive erosion of the dam without a major impact wave) are realistic possibilities and are therefore kept in the manuscript.
(2) The potential role of anthropogenic climate change: This is a complex topic and difficult to address in a short discussion section. Given the concerns of Reviewer #2, we have revised this section to provide clearer argumentation, and in particular, we have given more focus to the role of paraglaciation in linking the retreat of the glacier to the landslide/GLOF event.

We hope that we have carefully addressed all of your concerns and that our responses will fulfil your expectations. We will be happy to answer any further questions you might have.

Sincerely,

Guoxiong Zheng
(on behalf of the Author team)
Xinjiang Institute of Ecology and Geography
Chinese Academy of Sciences
zhengguoxiong17@mails.ucas.edu.cn

**Response to individual referees**

**Referee #1 (Dmitry Petrakov)**

**General comments**

High Mountain Asia is prone to cryosphere related hazards including glacier lake outburst floods (GLOFs) which are frequently recognized as major glacier hazard in the region. Bursts of large lakes could lead to floods traveling a few hundred kilometers, dramatic damage of infrastructure and life losses. However, lack of case studies lead poor understanding of triggering mechanisms and processes for the majority of occurred GLOFs. The authors point out that it is particularly true for GLOFs happened during monsoon period, when cloud-free satellite images are limited.

Assessment of GLOF long-term conditioning and short-term triggering factors as well as flood dynamics is highly relevant topic not just for Tibet but for whole High Mountain Asia. It is especially important now considering changing environment, including lake formation, expansion and drainage, changing land use patterns, growth of population and consequent implications for GLOF hazard and risk.

The manuscript provides high-quality case study with significant conclusions, important for many high-mountain regions with rapid glacier retreat. Remote sensing approach accompanied by eye-witness analysis and numerical modelling is definitely suitable tool for this research. Scenarios for r.avaflow simulations are fully reasonable and description is clear for readers. Authors have explained obtained results and related problems in details. Hypothesis to interpret sequence of events are interesting, however it is difficult to exclude single versions at this stage. Author's opinion on high and increasing role of lateral moraine collapses in GLOF triggering is interesting and logical despite still based on limited number of cases.

The manuscript is well written and free of technical errors, well structured, appropriate in length. References are adequate. All figures and the table are high-quality. The conclusions are clear and precise. The results obtained in this study are highly relevant to assess hazard of future GLOFs within regions with fast glacier retreat. Novelty of results is absolutely clear. I strongly recommend to publish the manuscript in the Cryosphere with very minor corrections.

**Response:** We thank the reviewer for these very detailed summaries and positive feedback on our manuscript. We have carefully revised the manuscript per the comments below.

**Specific comments.**
Line 81 – please indicate year of assessment.
**Response:** Done.

Fig.2 – it will better to identify date or season and year of photos.
**Response:** We agree and have added the date each photo was taken.

Fig.8 – it is not clear, why daily maximum temperature has been chosen as criteria instead of daily mean temperature. Please explain it.

**Response:** As in this case we are specifically interested in melt conditions prior to the landslide and GLOF events, so we focused on maximum daily temperatures. At this elevation and time of year, nights are frequently sub-freezing, and therefore actual melt conditions during the day may not be well reflected in mean temperature data.

Table 1 – *resolution of multispectral/panchromatic image. If so, for Ziyuan-3 02 and Gaofen-2 resolution of multispectral image is better than panchromatic?
**Response:** We thank the reviewer for pointing out this mistake and have corrected it.

Table 2 – a nice set of equations has been used for the volume assessments. My feeling that it will be better to include a couple of more regionally specified formulae developed by Yao et al. (2012), approach and statistics from (Veh et al., 2020, Supplementary) containing 24 Himalayan lakes which allows to refine volumetric assessments. Similarly, it will be better to include the relation between lake volume and peak discharge (Popov, 1991) was determined specifically for moraine lakes in Kazakhstan. Furthermore, in the Supplement there is no information about type of lakes have been used for equations. I might be wrong, but ice-dammed lakes were considered in some equations.
**Response:** We thank the reviewer for this excellent suggestion. We have considered these additional formulas and updated the corresponding figures and text in the revised manuscript. Also, we created new column 'Lake types' in Supplementary Table 1 (where specified).

You might also want to discuss a bit more about specific features of landslide-lake interactions and consequent impact on process chain.
**Response:** In light of the concerns of other reviewers, we have substantially revised the discussion section, which now contains more specific features of the landslide-lake interactions and consequent impact on the process chain.

Popov, N. 1991. Assessment of glacial debris flow hazard in the north Tien-Shan. In Proceedings of the Soviet-China-Japan Symposium and field workshop on natural disasters, 2–17 Sept. 1991, pp. 384–391.

Veh, G., Korup, O., Walz, A., 2020. Hazard from Himalayan glacier lake outburst floods. Proc. Natl. Acad. Sci. 117 (2), 907–912 http://sci-hub.tw/10.1073/pnas.1914898117

Yao, X., Liu, S., Sun, M., Wei, J., and Guo, W.: Volume calculation and analysis of the changes in moraine-dammed lakes in the north Himalaya: a case study of Longbasaba lake, J. Glaciol., 58, 753–760, 2012

**Anonymous Referee #2**

This paper describes in detail the development of a GLOF using a wide range of technical data, remote sensing and appropriate lake and GLOF modelling. It provides a nice regional example of how some GLOFs may develop and introduces some suggested lag effects that account for the difference in potential GLOF trigger and the actual flood. I think the paper is appropriate for The Cryosphere but some changes are suggested. These are listed below.

**Response:** We much appreciate the reviewer's positive feedback and have carefully revised the manuscript in line with the comments below.

In the introduction you could expand the references to include: Wilson et al (Wilson, R., Harrison, S., Reynolds, J., Hubbard, A., Glasser, N.F., Wündrich, O., Anacona, P.I., Mao, L. and Shannon, S., 2019. The 2015 Chileno Valley glacial lake outburst flood, Patagonia. Geomorphology, 332, pp.51-65). Also: Haritashya, U.K., Kargel, J.S., Shugar, D.H., Leonard, G.J., Strattman, K., Watson, C.S., Shean, D., Harrison, S., Mandli, K.T. and Regmi, D., 2018. Evolution and Controls of Large Glacial Lakes in the Nepal Himalaya. Remote Sensing, 10(5).

**Response:** Expanded.

Line 115. Figure 2. "Opposite side". Clarify this. Also reword the caption. What does 'front to back' refer to.

**Response:** We agree and have revised the 'opposite side' to 'opposite valley', and the 'front to back' to 'downstream to upstream of the lake'.

Line 167. Met data section. How important are climate and weather differences over 50km in this mountainous region? I would suggest they are significant (especially precipitation) given rainfall shadow effects. If so, the data may not be relevant for the site. The authors need at least to acknowledge these problems.

**Response:** Thanks for this excellent comment. We agree it is true that there is significant heterogeneity in climate and weather conditions in mountainous areas. However, considering the relatively sparse distribution of available weather stations in such areas and the limitations of resolution and accuracy of alternative reanalysis data, using station data from 50 km away is considered the most desirable option in this case. We are confident in the temperature synchronization between the lake and the meteorological site because of their very close elevation (4,489 vs 4,444 m asl). Possible differences could exist in precipitation as the reviewer noted, although two considerations give us confidence in using the station data. 1) We have analysed accumulated precipitation from GPM-IMERG product over a larger region of South-eastern Tibet, for June 2020. It shows a remarkable consistency between the station location of Lhari (244 mm) and the lake location (240 mm). A figure showing this general homogeneity in June 2020 precipitation between Lhari station and the lake location has been added to the Supplementary material (Supplementary Figure 1). 2) Through interviews with local people living in the downstream villages, we confirmed that there was indeed a continuous rainfall in the period before the GLOF, which is generally consistent with the trend recorded at the weather station. We have added a short discussion on this point and referred to the new Supplementary figure in the revised manuscript.

Line 300. The timing of the GLOF is not much warmer than the mean climate, especially compared with other periods of 2020 where it is both above and below the 1986-2015 climate mean. The potential link to precipitation is a bit more convincing but if heavy rainfall is the trigger then the use of a data set from 50km distance might not tell us much. The authors need to reflect on these caveats and complications. While temperature might show some regional synchronicity, it is not very likely that precipitation does.

**Response:** Over the 14 days immediately preceding the GLOF, temperatures were 1.6°C warmer than mean climate, peaking at 4.4°C on June 24th when temperatures exceeded 90th percentile values. To reflect the reviewer's concern, we have modified the sentence slightly to refer factually to "above-average temperatures" rather than the more subjective use of "unusually warm". We have also added further values to the sentences that follow. Note we are not claiming here, or elsewhere in the manuscript that above-average temperatures are a sufficient explanation for the events, but rather suggesting that a combination of drivers could have been involved.

See previous comment regarding precipitation. We are confident (based on Supplementary Figure 1, and eye-witness accounts) that the heavy precipitation in June 2020 measured by the station data in Lhari is consistent with a larger regional pattern of heavy rainfall, including over the Jinwuco catchment.

Line 309. Why do 2 volume equations not yield meaningful results? Make sure you haven't 'cherry-picked' the ones that do!

**Response:** We excluded these two because highly unrealistic lake volumes and mean depths were obtained (mean depth 5 m in case of Wang et al. (2012) equation and mean depth of 94 m in case of O'Connor et al. (2001)). These values lie outside the range mean +/- 1 STDEV from and we found these values disproportional to the actual size (area) of the lake. We have mentioned this in the revised version of the text.

Line 440. An alternative assessment might conclude that the landslide and the GLOF are unrelated given the lag times. The GLOF might just be a stochastic event? If so, the climate attribution might be premature (which I think is likely). At least the authors should discuss this issue.

**Response:** By checking more satellite images, we found that the previous interpretation of the radar images needed to be revisited. The unusual features shown in the 21 June radar images may be drift ice. A satellite scene acquired on 24 June (has been added to Fig. 3) clearly shows that the landslide had not yet occurred at this time. We therefore revise our interpretation, and assume with confidence that the landslide either occurred on the same day as the GLOF or at most two days before it. We would also note that other imagery immediately after the GLOF clearly shows the dirty, muddy colour of the lake resulting from the landslide. This implies that there is a very high (almost certain) likelihood of a direct causal relationship between them. The relevant sections of the discussion have been revised accordingly in the manuscript.

Line 450–457. This section is rather vague. Please discuss what you know and point out what you don't with more clarity.

**Response:** This section has been substantially revised based on new evidence found in additional satellite images. Please see revised Section 5.1.

Line 489. Attribution to climate change is very complicated. Climate warming probably caused the lake expansion (but this might be regional), although calving might also have played a role. The specific event could only have caused a GLOF if the lake already existed, but this does not represent a clear example of a climate change attribution. This section is relatively weak and does not consider what attribution really means here. I would also suggest that the issue of paraglaciation is extremely important, yet there is little consideration of this. Papers by Ballantyne and also by Knight could be cited and discussed.

**Response:** We fully appreciate the reviewer's concern and acknowledge the complexities involved in climate change attribution. A full attribution study would be a separate paper in itself (as shown by colleagues for the case of Lake Palcacocha recently), and that is not our primary interest here. However, at the same time, we feel it is important to touch on this topic in the discussion given its potential policy relevance. Especially given now our revised interpretation which almost certainly establishes the landslide as a direct trigger of the GLOF, we feel the potential link to anthropogenic climate change can be robustly argued, albeit not formally attributed. We have therefore heavily revised this section to provide a clearer line of argumentation, and now avoid the loaded term of "attribution", as for the reasons stated above, we cannot claim to go this far. In particular, we have given more focus to the issue of paraglaciation, and how it links the retreat of the glacier to the conditioning/triggering of the GLOF.

Line 508. Mention Wilson et al who also assessed GLOFs from moraine landslides.
**Response:** Expanded.

Finally, we are not told what the moraine dam is made of. How sensitive are the model equations to whether the moraine has an ice core or is composed of sediment. Also what type of sediment is likely, and how does this affect the model results? Are ice-cored moraines common in the area? At least you need to discuss these issues.

**Response:** We fully understand the reviewer's concerns. Unfortunately, we have no field or lab information (granulometry, geophysics, drill cores, etc.) on the moraine. However, the maximum erosion depth reconstructed from the pre- and post-event DEMs was 25 m. This, in our opinion, would rather indicate that the moraine has no substantial ice core, at least not in the area where the breach developed, because otherwise, such an ice core would likely have constrained the breach depth, and therefore also the degree of lowering of the lake level. The entrainment coefficient is calibrated in the way to yield plausible peak discharges, compared to empirically derived peak discharges. We have added these discussions in Section 5.2.

Overall, I think this is a nice paper with excellent figures and a comprehensive modelling exercise. It needs some changes but I think it could be published in this journal. There is a reservation about whether this paper is only of regional interest, but I think that they produce a well-argued and comprehensive case study.

**Response:** Many thanks. We hope we have now carefully addressed or clarified the issues raised above.

**Referee #3 (Ashim Sattar)**

The authors have analyzed the GLOF process chain of the 2020 GLOF at Jinwuco, Tibet. It is a robust study where the causes, impacts, and implications are well presented. The authors also point out that heavy precipitation can be an important trigger of GLOF. Meteorological conditions, especially extreme rain events, may partly melt ice in the moraine dam and weaken the moraine, or cause overfill and thermal and physical erosion of the moraine, or induce a mass movement into the lake. The effort is relevant from a scientific point of view and definitely falls under the scope of the journal. The manuscript is well structured. I highly recommend the paper and suggest some moderate changes. Hope my comments will be useful to the authors.
**Response:** We are very grateful to the reviewer for this great summary and positive feedback on our manuscript. We have carefully studied the following comments and revised the manuscript accordingly.

See below for the comments.

**Abstract:**
I would suggest including some results of the most likely scenario presented in the study.
**Response:** Despite the additional satellite imagery allowing us to substantially reduce the lag time of the landslide, we still cannot exclude either of the two scenarios based only on the existing evidence. We have included some results related to them in the revised manuscript.

Line 25-30 It is strange to have such a long lag time of 5-17 days between the landslide and the GLOF event. Overtopping waves are normally short-lived, and has an instantaneous or short-lived but high impact on the damming moraine. I would recommend rephrasing this sentence.
**Response:** After checking more satellite images, we realized that the previous interpretation of the radar images needed to be reconsidered. Imagery acquired from 24 June 2020 shows that the landslide had not yet occurred at this time, which suggests that the landslide may have occurred on the same day or at most two days or so before the dam failure. We have therefore removed the statements about the long lag time and revised the discussion accordingly.

As the lag time is over several days, overfill of the lake due to rainfall can be a cause for one of the scenarios in the study??
**Response:** As noted above, we now have more evidence suggesting that the interpretation of this long lag time was not reliable, while suggesting a closer relationship between the landslide and the GLOF.

**Introduction:**
I would recommend focusing on studies that reconstructed previous GLOF events
(e.g. Westoby, M.J., Glasser, N.F., Hambrey, M.J., Brasington, J., Reynolds, J.M. and Hassan, M.A., 2014. Reconstructing historic Glacial Lake Outburst Floods through numerical modelling and geomorphological assessment: Extreme events in the Himalaya. Earth Surface Processes and Landforms, 39(12), pp.1675-1692. Nie, Y., Liu, W., Liu, Q., Hu, X. and Westoby, M.J., 2020. Reconstructing the Chongbaxia Tsho glacial lake outburst flood in the Eastern Himalaya: Evolution, process and impacts. Geomorphology, 370, p.107393. Westoby, M.J., Brasington, J., Glasser, N.F., Hambrey, M.J., Reynolds, J.M., Hassan, M.A. and

Lowe, A., 2015. Numerical modelling of glacial lake outburst floods using physically based dam-breach models. Earth Surface Dynamics, 3(1), pp.171-199.

Majeed, U., Rashid, I., Sattar, A., Allen, S., Stoffel, M., Nüsser, M. and Schmidt, S., 2021. Recession of Gya Glacier and the 2014 glacial lake outburst flood in the Trans-Himalayan region of Ladakh, India. Science of The Total Environment, 756, p.144008.)

**Response:** Thanks for this suggestion. We have now referred to these studies in the Introduction.

Last para:

I would suggest introducing the event further by adding one or two sentences to emphasize the presented case study.

**Response:** We agree and have added the following introductory sentences at the beginning of the paragraph: "During the onset of the 2020 monsoon, under the influence of warm temperatures and heavy rainfall, a GLOF occurred on 26 June in Tibet, China, causing severe damage to infrastructure and social impacts in downstream areas."

**Materials and methods:**

Section 3.4

The values of breach and lake characteristics fit better in the Result section. It remains unclear how the volume of the lake was empirically determined. Was it from a particular empirical equation or the average of all? Sorry if I missed it.

**Response:** We prefer to keep these values mentioned already in the materials and methods section because they were used as input data to fuel our volume/depth/peak discharge/breach time estimations, which are then presented in results section 4.4. Instead of determining a lake volume, we provide a range of (likely) volumes, considering different empirical equations. We believe that providing a range of solutions is relevant when actual lake bathymetry is not known (and we cannot do much about it). For the modelling purposes, we chose a value of 13.9 $Mm^3$, which is slightly more conservative than mean (15.9 $Mm^3$) and median (14.4 $Mm^3$) estimates (these revised estimates consider more equations and are slightly higher than those in the previous version of the manuscript).

Section 3.5

The flow propagation is analyzed up to 1.2 km downstream. I note that validation of the flow is difficult in the downstream part of the channel. However, it can be assumed that if the initial conditions of reconstruction (trigger and breach) were validated (which the authors have done a good job), the downstream propagation more or less will represent the flow. E.g. inundations at places shown in Fig 12 and 13, table 3, etc can be helpful here to evaluate the modeled inundation.

**Response:** This is a very important comment. Initially, we tried to do some simulations down to the main valley but finally decided not to include this type of simulation into the paper for the reason that the quality of the DEM is not good enough. The bottom of the main valley is very wide, meaning that even a moderate error in the DEM means that the flow takes a completely different course than it did in reality. This issue would make such a simulation a purely technical exercise without any scientific value, as evaluation (e.g. against available photos and videos) would be meaningless.

At places in the section, results are mentioned e.g. Line 216, Line 218, etc. I would suggest shifting it to the results section.

**Response:** Done.

**Results:**

Section 4.3 Line 310–313: As mentioned earlier the lag time between the landslide and GLOF is several days. This can be discussed.

**Response:** As noted in the responses above, the reanalyses of additional satellite images has now considerably reduced the lag time between the landslide and GLOF, suggesting a closer relationship between these two events. Text and discussion have been revised accordingly.

Section 4.4 Line 319–321 How was the reconstructed lake volume of 13.9 M m$^3$ calculated? This can be explained in the methodology and result section 4.2 or 4.4.

**Response:** Out of the range of estimated lake volumes, we chose this one as a slightly more conservative value than the revised mean (15.9 Mm$^3$) and median (14.4 Mm$^3$) estimates (these revised estimates consider more equations and are slightly higher than those in the previous version of the manuscript). We have mentioned this in the revised version of the manuscript.

Section 4.5 Line 343 The peak discharge reaches 10,900 m$^3$/s. It does not reflect in the hydrograph in Fig.10.

**Response:** This peak discharge was computed at a higher temporal resolution and thus not visible in Fig. 10. The relevant explanation has been added to the manuscript.

**Discussion:**

The authors have done a great job here discussing the uncertainties in detail. This also answers some of my questions above. I agree that reconstructing GLOF is associated with a lot of uncertainties, lack of evidence, the complexity of the triggering process, etc. Evaluating multiple scenarios to reconstruct the process chains can however be useful to understand the event and not rule out the various possibilities.

**Response:** We thank and fully agree with this comment. Considering the complexity of the GLOF process chain and the difficulty of capturing the actual situation on-site, it is reasonable and useful to consider multiple scenarios when reconstructing such events. Although the existing evidence we have points more to scenario A, we could not completely exclude scenario B, so both scenarios have been kept in the manuscript.

**Figures:**

Fig.1 Township border and river colors can be changed.

**Response:** We have modified the colour of the township border to make it more distinguishable from the river.

Fig.2 Caption – To avoid confusion, front to back can be replaced with downstream to upstream. Similarly left to right or right to left can be represented in terms of directions with respect to north.

**Response:** We have adopted this useful suggestion.

Fig.3 recheck the scale of Fig. a; dates in a and b is not visible.

**Response:** We have rechecked and added backgrounds to the dates to make them more visible.

Fig.10 $H_{p1}$, $H_{p2}$, $Q_{p1}$, $Q_{p2}$…etc can be mentioned in the caption. The discharge represented on a negative scale? Sorry if I missed anything.

**Response:** H means flow height, and Q means discharge. The subscript P1 refers to phase 1 (rock), P2 to phase 2 (non-existent in this case), and P3 to phase 3 (water). Fluid discharge and flow height are plotted in the negative direction for better readability. We have added these pieces of information to the figure caption.

Table 1 Line 140 – Date or Data?

**Response:** It is 'Data' and has been corrected.

Overall a very nice study that includes remote sensing and complex modeling to understand GLOF events. I highly recommend this paper with moderate revisions

**Response:** We thank the referee very much again for the constructive input which has greatly contributed to the refinement of our manuscript.